# A framework for dissecting affinities of multidrug efflux transporter AcrB to fluoroquinolones

Julia Vergalli[1,6], Hugo Chauvet[2,6], Francesco Oliva [3,6], Jelena Pajović[2,4], Giuliano Malloci [3], Attilio Vittorio Vargiu [3], Matthieu Réfrégiers [2,5], Paolo Ruggerone [3] & Jean-Marie Pagès [1✉]

Sufficient concentration of antibiotics close to their target is key for antimicrobial action. Among the tools exploited by bacteria to reduce the internal concentration of antibiotics, multidrug efflux pumps stand out for their ability to capture and expel many unrelated compounds out of the cell. Determining the specificities and efflux efficiency of these pumps towards their substrates would provide quantitative insights into the development of antibacterial strategies. In this light, we developed a competition efflux assay on whole cells, that allows measuring the efficacy of extrusion of clinically used quinolones in populations and individual bacteria. Experiments reveal the efficient competitive action of some quinolones that restore an active concentration of other fluoroquinolones. Computational methods show how quinolones interact with the multidrug efflux transporter AcrB. Combining experiments and computations unveils a key molecular mechanism acting in vivo to detoxify bacterial cells. The developed assay can be generalized to the study of other efflux pumps.

[1] UMR_MD1, U-1261, Aix-Marseille Univ, INSERM, IRBA, MCT, Marseille, France. [2] DISCO beamline, Synchrotron Soleil, Saint-Aubin, France. [3] Department of Physics, University of Cagliari, 09042 Monserrato, (CA), Italy. [4] Present address: University of Belgrade, Faculty of Physics, 11001 Belgrade, Serbia. [5] Present address: Centre de Biophysique Moléculaire, CNRS UPR4301, Rue Charles Sadron, Orléans, France. [6] These authors contributed equally: Julia Vergalli, Hugo Chauvet, Francesco Oliva. ✉email: Jean-Marie.PAGES@univ-amu.fr

In Gram-negative bacteria, the membrane transporters including porins and efflux pumps have been extensively studied for their involvement in antibiotic translocation[1–3]. However, there is still a serious gap in our understanding of antibiotic transport across the envelope of living bacteria[3,4], especially because of the sophisticated regulation-structure-function of efflux pumps.

Fluoroquinolones (FQs) belong to the quinolone family of antibiotics that are widely used to treat human and animal infections[5]. Bacterial resistance to this class of compounds implies several strategies: mutational alteration of the drug targets and target masking, reduction of envelope permeability and expression of active efflux pumps[2]. These latter two membrane-associated mechanisms of resistance, strongly control the intracellular accumulation of drugs and decrease their concentration close to their targets[1]. Unfortunately, complete understanding of these mechanisms has not been convincingly reached until now.

Recently, by using a method based on fluorimetric assays[6], a collection of FQs has been studied to determine their accumulation level in *Escherichia coli* expressing or not AcrB, the major and most-studied multidrug efflux transporter of the Resistance Nodulation cell Division super-family[2]. The Structure Intracellular Concentration Activity Relationship (SICAR) index has been assayed in addition to drug activity determined as dose for early killing, coupled with a detailed molecular-level view obtained through complementary computational techniques. The SICAR concept has been clearly validated as a powerful tool for comparing FQs properties and defining their behaviour with respect to efflux pumps expression[7].

As previously demonstrated, by increasing the amount of efflux substrate, a partial saturation of the pump transport capability was observed[8]. Starting from this observation, which opened the possibility to study drug efflux susceptibilities of an entire antibiotic family, we developed an AcrB-specific competition assay between quinolones in a multi-drug-resistant clinical isolate of *Klebsiella aerogenes*[9].

By comparing the intracellular accumulation levels of different fluoroquinolones resulting from a set of fluoroquinolone-quinolone combinations, our method aims at assessing their relative affinities to AcrB. The results obtained highlight the substrate-AcrB interactions with relative affinities varying from 5 to 500 among the quinolone family. These data are rationalized by means of a hierarchically built computational protocol, demonstrating the intriguing relationship between specific interactions with AcrB and the resulting activity of key compounds. The obtained information unveils possible new strategies based on the combination of antibiotics involved in both competitive inhibition and direct antibacterial action.

## Results

We considered four fluorescent FQs, ciprofloxacin (CIP), enrofloxacin (ENR), fleroxacin (FLE) and norfloxacin (NOR) and three non-fluorescent quinolones, nalidixic acid (NAL), rosoxacin (ROS) and sparfloxacin (SPA) (Supplementary Fig. 1). Quinolones were hypothesized to compete with FQs in binding to the AcrB efflux transporter, thus altering FQs intracellular concentration and activity. For this reason, hereafter we docket non-fluorescent quinolones as competitors (CPTs) to distinguish them from FQs. The considered compounds, including drugs currently used for human (CIP, NOR) and veterinary (ENR) therapy, have been previously studied in bacteria expressing different levels of AcrB[2], enabling the determination of FQs SICAR indexes (Supplementary Fig. 1).

The study focused on two isogenic strains: EA27, a well-characterized clinical isolate over-expressing AcrAB (the complex formed between the Resistance Nodulation cell Division efflux transporter AcrB and the adaptor protein AcrA), and containing a quinolone resistance-determining regions (QRDR) mutation (see Methods), and EA294, its derivative strain devoid of AcrB.

**FQs and CPTs had high activities with various susceptibilities to AcrB efflux on the multidrug resistant clinical isolate EA27.** Among the FQs, CIP, ENR and FLE were similarly active with minimum inhibitory concentrations (MICs) of 64-128/4-8 µM on EA27/EA294, and NOR was less effective with MICs of 512/64 µM on EA27/EA294 (Table 1). FQs activities were 3- (NOR) 4- (CIP and FLE) and 5-fold (ENR) decreased on EA294, indicating a medium to high efflux susceptibility in the AcrAB overexpressing strain. Regarding the CPTs, NAL and ROS had very low activities on EA27 (4- to 7-fold decreased compared to FQs) with MICs greater than or equal to 8192 µM, while SPA showed a similar activity than CIP, ENR and FLE. NAL and ROS maintained high MICs in the efflux deficient strain, indicating a moderate efflux susceptibility in EA27. On the contrary, SPA was more susceptible to efflux with a 3-fold decrease in EA294.

The effect of FQs and CPTs on metabolism was determined using the resazurin-based bacterial viability assay (see Methods). Supplementary Fig. 2 shows the metabolic activity (expressed in % and determined from the percentage resazurin reduction) of EA27 and EA294 incubated in the presence of increasing concentrations of FQs and CPTs. FQs first effects on EA27 metabolic activity were induced by 4/8 µM CIP/ENR and 32/64 µM FLE/NOR. ENR was the FQ with the activity most susceptible to efflux (3-fold decreased in EA27 compared to EA294). As suggested by MICs data, SPA activity, which was similar to FQs activities, was higher than ROS and NAL on both strains with first effects on metabolic activity induced by 32/4 µM of SPA compared to 128/8 µM of ROS and 512/128 µM of NAL on EA27/EA294. However, ROS seemed more susceptible to efflux in EA27 compared to what revealed by MIC data since its activity was 4-fold decreased in EA27 compared to EA294.

**Different CPTs doses were needed to restore FQs accumulation and saturation of efflux pump.** The internal accumulation of FQs was monitored in the EA27 strain in the absence or in the presence of the CPTs. As a control, we used carbonyl cyanide *m*-chlorophenyl hydrazone (CCCP), which is known to inactivate AcrB by dissipating the transmembrane potential needed by the efflux pump. Bacterial cells were incubated with a fixed concentration of a selected FQ, with and without CCCP, at increasing concentrations of different CPTs. The intracellular FQ accumulation was measured at 20 minutes (Fig. 1), which corresponds to the FQ accumulation plateau[7]. The three CPTs induced an increase in the amount of the four FQs inside the cells. The CPTs concentrations required to increase the FQs accumulation to the

---

**Table 1 FQs and CPTs activities. MICs (µM) of FQs and CPTs measured on the multidrug resistant clinical isolate EA27 and the efflux-deficient derivative strain EA294. Three independent assays were performed.**

|      | EA27    | EA294 |
|------|---------|-------|
| CIP  | 64      | 4     |
| ENR  | 64–128  | 4     |
| FLE  | 128     | 8     |
| NOR  | 512     | 64    |
| NAL  | 8192    | 4096  |
| ROS  | ≥8192   | 2048  |
| SPA  | 128     | 16    |

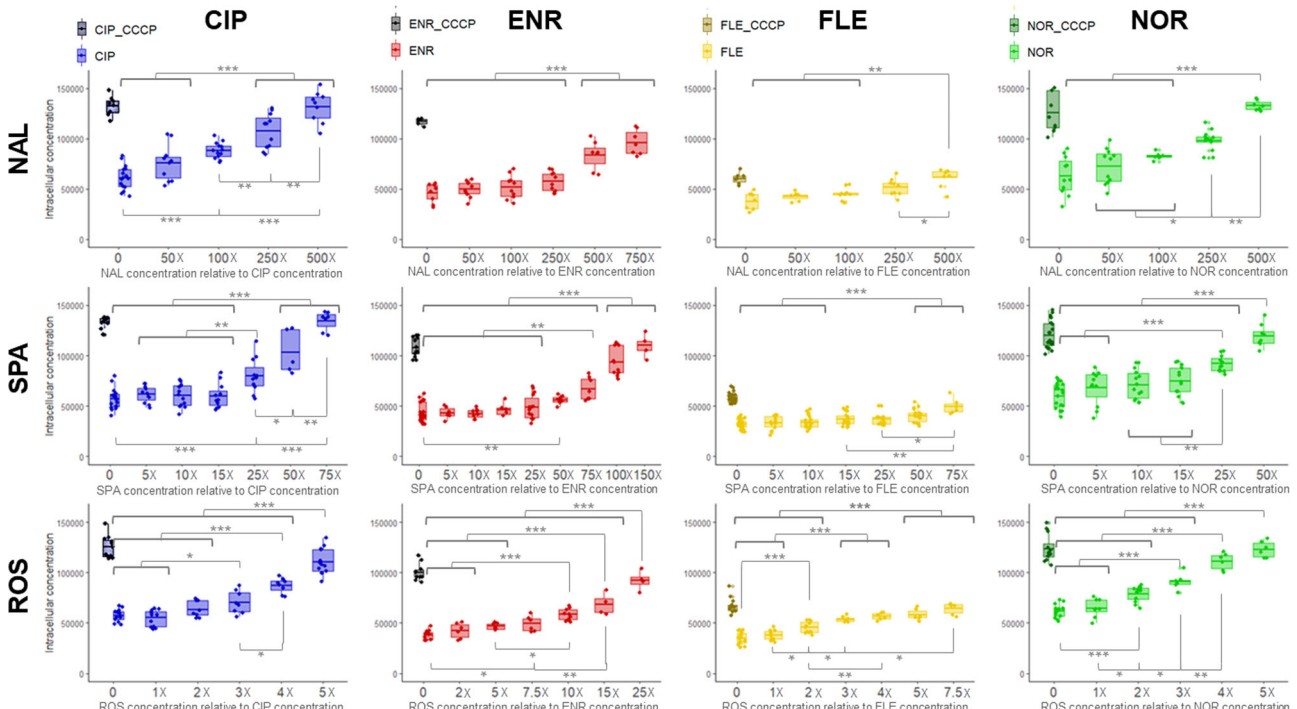

**Fig. 1 CPTs dose effect on FQs intracellular accumulation in bacterial population.** Boxplot of intracellular concentrations of FQs (CIP, ENR, FLE, NOR, in molecules.cell$^{-1}$) accumulated in EA27 incubated without or with CCCP, or with increasing concentrations of CPTs (NAL SPA, ROS). The concentrations used during incubations were 8 µg.ml$^{-1}$ FQ, 20 µM CCCP, X molar equivalent of CPTs relative to the co-incubated FQ concentration. Bacterial suspensions were sampled at 20 min and intracellular concentrations of FQ were determined using spectrofluorimetry (see 'Methods'). Results were obtained from $n = 4, 4, 4, 4, 4, 4, 6, 4, 3, 3, 3, 3$ independent experiments performed in triplicate. ANOVAs with Tukey's posthoc tests ($F_{(4,53)} = 59.4$, $P < 2 \, 10^{-16}$; $F_{(6,49)} = 30.5$, $P = 5.9 \, 10^{-15}$; $F_{(4,31)} = 25.7$, $P = 1.84 \, 10^{-9}$; $F_{(4,42)} = 48.4$, $P = 3.4 \, 10^{-15}$; $F_{(6,69)} = 53.3$, $P < 2 \, 10^{-16}$; $F_{(8,76)} = 60.7$, $P < 2 \, 10^{-16}$; $F_{(6,93)} = 9.7$, $P = 2.8 \, 10^{-8}$; $F_{(5,70)} = 38$, $P < 2 \, 10^{-16}$; $F_{(5,57)} = 72.4$, $P < 2 \, 10^{-16}$; $F_{(6,41)} = 45$, $P < 2 \, 10^{-16}$; $F_{(6,54)} = 45.3$, $P < 2 \, 10^{-16}$. $F_{(5,45)} = 91.9$, $P < 2 \, 10^{-16}$ and $\eta^2 = 0.8, 0.75, 0.45, 0.45, 0.79, 0.85, 0.38, 0.66, 0.85, 0.86, 0.81, 0.89$ for the pairs CIP-NAL, ENR-NAL, FLE-NAL, NOR-NAL, CIP-SPA, ENR-SPA, FLE-SPA, NOR-SPA, CIP-ROS, ENR-ROS, FLE-ROS and NOR-ROS) were used to determine differences of FQ accumulation (***$P < 0.001$; **$P < 0.01$; *$P < 0.05$).

level obtained with CCCP were the highest with NAL and the lowest with ROS. For NOR accumulation, NAL 500X, SPA 50X, or ROS 5X were needed. Although the same trend among CPTs was observed with the three other FQs (NAL > SPA > ROS), some differences were noted regarding the effect of CPT. For instance, a higher CPT concentration was needed for increasing ENR accumulation compared to the three other FQs. Interestingly, concerning the capability to block efflux activity, 150 µM ROS (5X) induced an increase of intracellular NOR like that observed with 20 µM of CCCP. This indicates that the effects of the two blocking mechanisms - energy *vs.* competition - is of the same order of magnitude under our assay conditions.

In order to validate the effects of CPTs in EA27, accumulation of NOR and ENR co-incubated with ROS or NAL were measured in the no-efflux isogenic strain EA294 (Supplementary Fig. 3a). Similar levels of accumulation of both FQs were obtained in EA294 in all tested conditions. These results validate the involvement of efflux pump saturation by the CPT-FQ pairs in the increased FQs accumulation observed in EA27.

**Chloramphenicol and erythromycin were as weak competitors as NAL.** Considering the data obtained with quinolones competitors, it is important to control what happens using other antibiotic families. To this aim, we have selected chloramphenicol and erythromycin and performed the assay in similar conditions (by using increasing concentration of putative competitors). Although these two molecules are efflux substrates, huge concentrations were required to obtain an increase of NOR and ENR

accumulation (Supplementary Fig. 4). Indeed, chloramphenicol and erythromycin 5X were not enough to increase the NOR accumulation as ROS 5X did. Moreover, erythromycin competition induced similar effects on NOR and ENR accumulation than NAL. So, although belonging to the quinolone family, NAL did not seem structurally close enough to the studied FQs compared to ROS to compete with them better than chloramphenicol or erythromycin. Note that NOR and ENR accumulation levels were similar in the no-efflux strain EA294 whatever the concentrations of chloramphenicol or erythromycin (Supplementary Fig. 3b). This data validates the observations from Fig. 1 indicating the importance of structural similarity between the various substrates.

**NOR-ROS was the best pair to saturate efflux in EA27.** The change in FQ accumulation in EA27 due to CPTs was measured with respect to the value obtained without adding them to the batch in the absence and in the presence of CCCP (corresponding to 0% and 100%, respectively, see Fig. 2a). This allows both to correct for variations due to different FQs penetration rates (SICAR$_{IN}$ as previously described in Ref. [2]), and to compare the competition ratio of the different CPT-FQ pairs inside AcrB. FQs accumulation percentages and the corresponding trend curves were plotted as a function of CPT concentrations (Fig. 2a). The CPTs Concentrations needed to Increase FQ Accumulation by 50% compared to CCCP (CCIA$_{CPT}$) were extracted from the trend curves (Fig. 2b): ROS was the best CPT (CCIA$_{ROS}$ was 6 times lower than the CCIA$_{SPA}$ and 70 times lower than the CCIA$_{NAL}$), NAL the less efficient. Interestingly, the CCIA$_{SPA}$ of

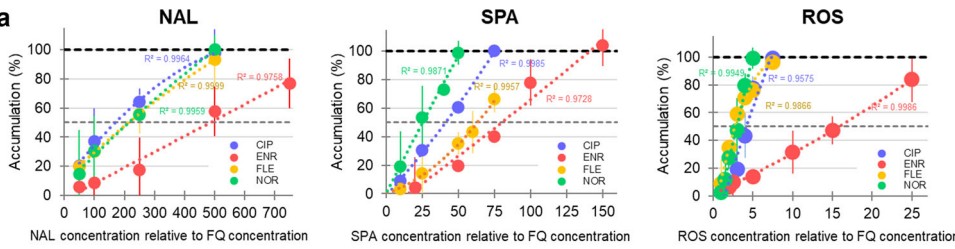

**Fig. 2 CPTs concentrations needed to increase FQ accumulation by 50%. a** Increasing FQ accumulation rate (%) as a function of the co-incubated CPT concentrations. FQs accumulation % (± SD) were calculated from the data shown in Fig. 1 where the FQ accumulation measured without CPT in the absence and in the presence of CCCP correspond to 0% and 100%, respectively. **b** CPTs Concentrations needed to Increase Accumulation of FQ by 50% compared to CCCP (CCIA$_{50}$), calculated from the trend curve (degree 2 polynomial) plotted in **a**.

each FQ remarkably differed, even if the CCIA$_{SPA}$ of ENR maintained the highest value, suggesting versatility during competition (possibly due to various parameters such as binding affinity, site availability, steric hindrance, etc.). Importantly, the ranking of FQs CCIA close to 100% (NOR ≤ CIP ≤ FLE < ENR) was conserved for the three CPTs.

According to these results, ROS competed better with FQs than NAL for AcrB. Note that the effect of ROS on NOR accumulation was time dependent (Supplementary Fig. 5). In addition, ENR and NOR were respectively the best and the worst AcrB substrate among FQs, inducing weak or high competition by CPTs for the efflux pump. Thus, NOR-ROS and ENR-NAL were respectively the best and the worst pair to induce an increase of FQ accumulation.

**NOR-ROS was the best pair for efflux pump saturation in various strains and species of multidrug resistant clinical isolates.** In order to ensure that the trend of efflux competition observed in EA27 was not restricted to this strain/species, efflux competition assays were performed in further multidrug resistant clinical isolates of *K. aerogenes* (EA3 and EA117), *K. pneumoniae* (KP55) and *E. coli* (ARS100 and ARS108). ROS was the best CPT to restore NOR or ENR accumulation in the various isolates, while high concentrations of NAL were needed to obtain the same level (Supplementary Fig. 6). Indeed, ROS 5X/25X restored the highest accumulation levels of NOR/ENR in all the strains but in KP55. Inversely, NAL 5X did not increase NOR or ENR accumulation whatever the strain. Interestingly, NAL 100X allowed to restore NOR accumulation in all the strains except in EA27. In the same way, NAL 100X was sufficient to restore ENR accumulation in EA3. However, NAL 500X was needed in KP55, ARS100 and ARS108, and NAL 500X was not enough in EA117.

**CPTs doses ranging from 5 to 500X affected NOR accumulation in individual bacterial cells.** The effect of the three CPTs on NOR accumulation was monitored in individual cells by microspectrofluorimetry[6]. Figure 3 shows the NOR accumulation in cells at two concentrations of CPT (selected from data of Figs. 1 and 2). A level like that obtained with CCCP was observed with ROS 5X, SPA 50X and NAL 500X. In contrast, SPA 5X and NAL 50X did not increase the level of NOR accumulation. Interestingly, the

highly variable NOR accumulations in cells co-incubated with the CPTs reflect the heterogeneity of bacterial population responses according to various physiological states. Moreover, this variability appeared greater than that detected in cells co-incubated with CCCP, probably due to different parameters involved (pump competition vs. energy dissipation). All these results sustain that the NOR-ROS pair is the best combination allowing the maximum FQ accumulation with the lowest CPT concentration.

**Cooperative cell killing rates due to the NOR-ROS combination.** To investigate the consequence of increasing NOR accumulation (due to ROS co-incubation) on bacterial susceptibility, the cell killing rates were measured in assays with NOR used at 1/32 of the MIC (for which no antibacterial effect was detected) in the absence or in the presence of ROS at 5X concentration (equivalent to 1/100 of the ROS MIC). Figure 4 shows cell killing rate (red points) and NOR accumulation (boxplots) measured in EA27 at 30 (Fig. 4a) and 60 minutes (Fig. 4b) in the absence or in the presence of CCCP or ROS 5X. There was no or a low cell killing rate during incubations of NOR 16 µM alone (2 and 1% at 30 and 60 minutes, respectively).

ROS co-incubation increased NOR accumulation at 30 and 60 minutes, leading to a corresponding rise of the bacterial killing rates by 23 and 39%, respectively (approximately twice than those detected with ROS alone). Note that target mutations present in the clinical strain reduces the NOR antibacterial action despite an increased internal concentration and mitigates the effect of compounds mixing on antibiotic activity.

**Molecular docking calculations identify preferred binding modes of FQs and CPTs at the deep binding pocket of AcrB.** According to the experimental findings described above, the efflux of AcrB substrates such as FLE, NOR, CIP and ENR, are competitively inhibited by the three CPTs with different efficiency. In silico experiments were therefore performed to provide an atomistic view of the interactions between FQs/CPTs and AcrB. Following previous investigations[2,10], a systematic ensemble docking campaign (Fig. 5) was carried out (see Methods Section). We focused on the deep binding pocket of the Tight monomer (DP$_T$), a prominent recognition and binding site of

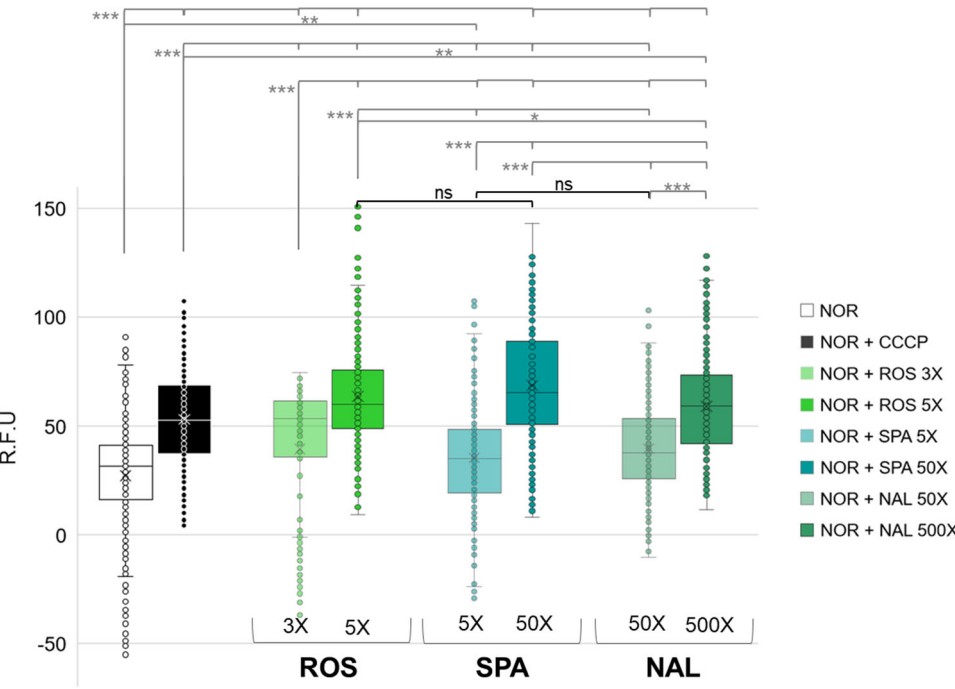

**Fig. 3 CPTs effect on NOR accumulation in individual cells.** Accumulation of NOR incubated without and with 2 concentrations of CPTs measured by microspectrofluorimetry in individual cells. Data were obtained from incubation of EA27 with 8 µg.ml$^{-1}$ of NOR without or with 20 µM of CCCP or with ROS 3X and 5X, SPA 5X and 50X, NAL 50X and 500X. CPTs concentrations are molar equivalent relative to the co-incubated FQ concentration. Bacterial suspensions were sampled at 20 min and pellets were resuspended in buffer and analyzed by deep–ultraviolet fluorescence imaging. Images were analyzed and corrected to compare the various conditions (see 'Methods'). The corrected relative unit of fluorescence (R.F.U) of NOR accumulation in individual cells was presented. Results were from at least 6 different microscopy fields per condition, each containing at least 20 bacteria, and this experiment was independently repeated two times ($n = 312, 423, 200, 288, 336, 205, 321, 274$ bacterial cells incubated with NOR, NOR + CCCP, NOR + ROS 3X, NOR + ROS 5X, NOR + SPA 5X, NOR + SPA 50X, NOR + NAL 50X, NOR + NAL 500X, respectively). Dunn tests were used to determine differences of NOR accumulation ($\eta^2 = 0.24$). ***$P < 0.001$; **$P < 0.01$; *$P < 0.05$; ns not significantly different.

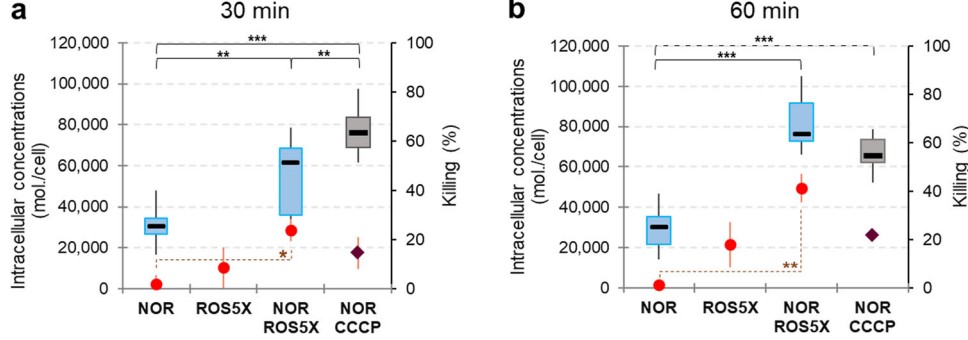

**Fig. 4 Cooperative killing rates of the ROS–NOR combination.** NOR accumulation (molecules.cell$^{-1}$) and cell killing rates (red points) measured during incubation of EA27 with NOR at 16 µM, without or with 20 µM CCCP, or with ROS 5X. Bacterial suspensions were sampled at 30 (**a**) and 60 minutes (**b**) for spectrofluorimetric measurement of NOR accumulation (boxplot) and for CFU determination. Cell killing rates were calculated from CFU and represented as means ± SD ($n = 3$ independent experiments). Kruskal-Wallis with Dunn tests were used to determine differences of NOR accumulation ($n = 3$ independent experiments) at 30 minutes ($\eta^2 = 0.79$) and 60 minutes ($\eta^2 = 0.42$). ANOVAs with Tukey post hoc tests were used to determine differences of killing rates at 30 min ($F(3,8) = 4.9$, $P = 0.0327$) and 60 minutes ($F(3,8) = 7.33$, $P = 0.011$). ***$P < 0.001$; **$P < 0.01$; *$P < 0.05$.

AcrB, believed to interact with every substrate during the extrusion process[11–13]. The portion of the DP$_T$ comprising multiple phenylalanine residues (F136, F628, F610, F615, F628), known as the "hydrophobic trap" (HT), is a crucial recognition site for AcrB inhibitors, and was also shown to interact with several substrates[12,14]. In Fig. 5b, c we analyse docking results in terms of the spatial distribution and binding affinity of all binding modes. Most poses are in the HT, which appears to be the highest affinity site in all cases. A few poses of all compounds but ROS and ENR are in the region between the DP$_T$ and the so-called exit gate. As

shown in Fig. 5b, NAL and ROS display the poses with the lowest and highest affinity to the HT, respectively. ROS, SPA and NAL show respectively an average binding affinity slightly higher ($-9.2$ kcal.mol$^{-1}$), similar ($-8.6$ kcal.mol$^{-1}$), and slightly lower ($-7.4$ kcal.mol$^{-1}$) than the average binding affinity estimated for the FQs ($-8.3$ kcal.mol$^{-1}$; see Fig. 6). Although qualitative, this trend is in line with the measured ability of the three CPTs to increase FQs concentration inside cells. An in-depth analysis of the contacts between the quinolones and the protein, reported in the Supplementary Note 1 and Supplementary Figs. 7, 8 and 9,

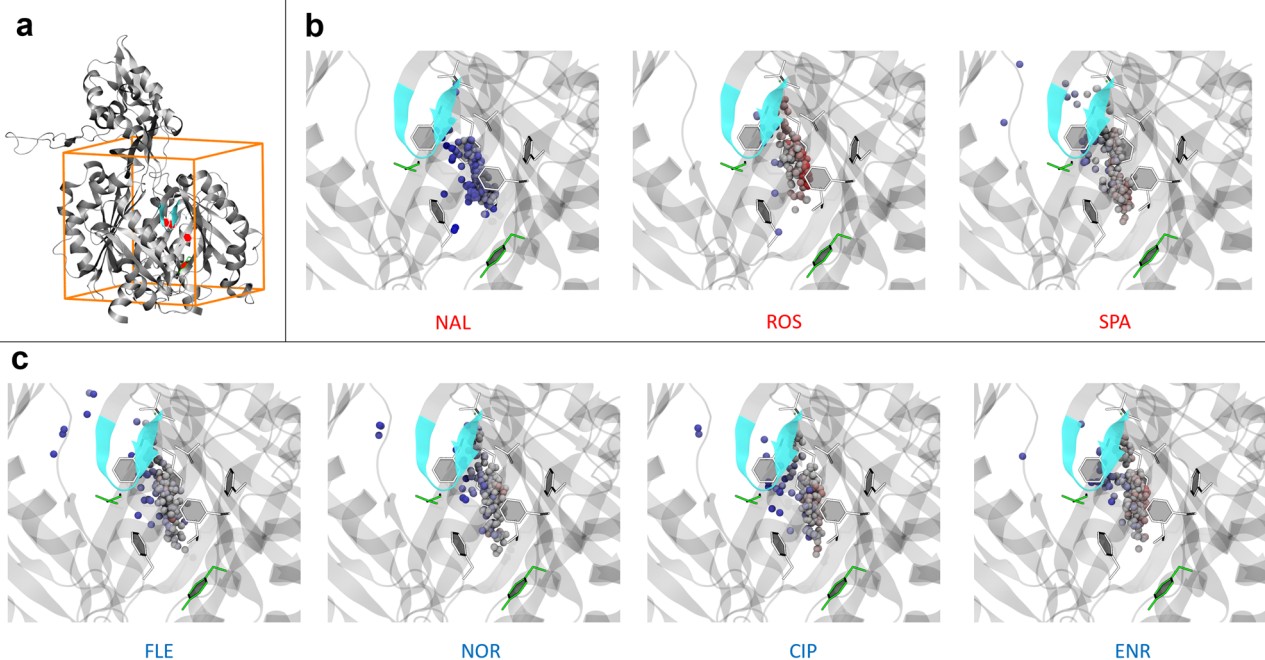

**Fig. 5 Ensemble docking of CPTs and FQs targeting the deep binding pocket of AcrB. a** Visualization of the adopted docking volume. The T monomer of AcrB is shown in grey ribbons. **b**, **c** Distribution of the center of mass of the docking poses. Licorice residues identify the HT while the light blue ribbon represents the switch loop. Colour code for the binding affinity: red and blue coloured spheres represent more and less "stable" poses, respectively, ranging from −10.9 to −6.1 kcal.mol$^{-1}$.

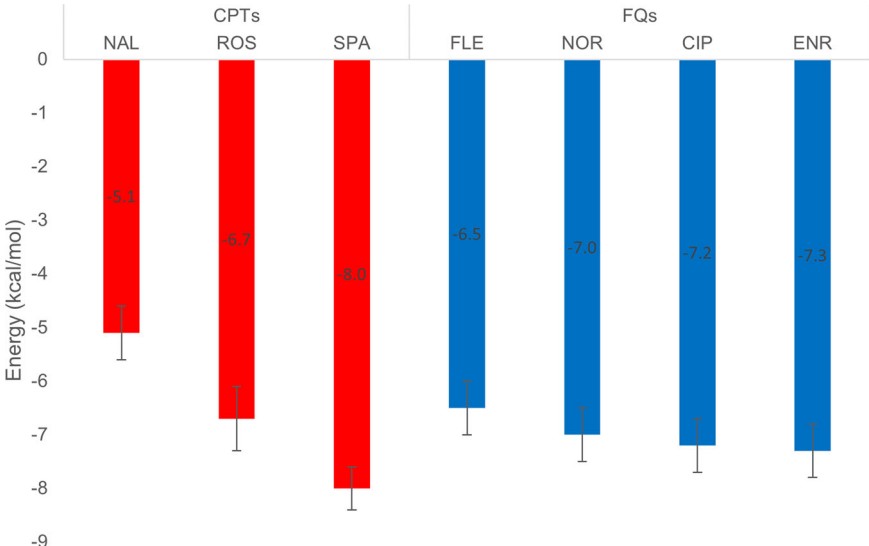

**Fig. 6 Average binding affinities from ensemble docking calculations.** Average binding affinities (±SD, $n = 180$) for each ligand as obtained through molecular docking performed using an ensemble of 18 X-ray conformations of AcrB.

agrees with experimental results. We found some similarities between NOR and ROS, while NAL behaves differently, interacting mainly with residues F136, V139, G179, I277, Y327, F610, F628. These results are again consistent with the competition assay showing NAL as the worst competitor and ROS as the best one.

**Molecular Dynamics simulations of AcrB in complex with FQs and CPTs suggest a possible mechanism of efflux competition.** In view of the experimental data, we focused our attention on NAL, ROS and NOR. Out of the 180 docking poses generated for each compound, those with the highest binding affinities (−10.9, −9.4, and −8.5 kcal.mol$^{-1}$ for ROS, NOR, and NAL respectively) were subjected to 10 independent molecular dynamics (MD) simulations each. Overall, during the dynamics, NOR was remarkably more stable than the two CPTs (Table 2). The latter undergo a change in their orientation with respect to the initial docking pose and after a displacement towards the HT, they wander within this region during the equilibrium trajectory. The free energies of binding ⊗G estimated from each MD trajectory with the molecular mechanics generalized Born surface area method (Table 3), confirm that NOR has a higher average affinity than the two CPTs for the transporter (in line with docking

**Table 2 RMSD values of NOR, ROS and NAL.** RMSD values (in Å) for the 10 replicas of the three quinolones NOR, ROS and NAL were obtained using the first frame of the trajectory as reference.

| | NOR | | | ROS | | | NAL | | |
|---|---|---|---|---|---|---|---|---|---|
| Replica | Avg. RMSD | Std. Dev. | Last RMSD value | Avg. RMSD | Std. Dev. | Last RMSD value | Avg. RMSD | Std. Dev. | Last RMSD value |
| 1 | 6.5 | 1.8 | 7.0 | 5.2 | 1.0 | 4.1 | 4.5 | 2.3 | 5.7 |
| 2 | 6.4 | 1.9 | 9.0 | 8.0 | 2.4 | 8.2 | 1.6 | 0.6 | 2.5 |
| 3 | 7.3 | 1.7 | 8.2 | 8.5 | 1.8 | 6.0 | 1.4 | 0.6 | 1.4 |
| 4 | 4.7 | 1.1 | 4.3 | 5.0 | 1.2 | 4.2 | 3.1 | 2.7 | 7.0 |
| 5 | 7.4 | 1.8 | 8.5 | 8.2 | 5.1 | 14.6 | 1.7 | 0.6 | 2.1 |
| 6 | 5.2 | 1.5 | 4.1 | 5.0 | 0.8 | 4.7 | 3.0 | 2.7 | 7.5 |
| 7 | 5.6 | 1.4 | 7.9 | 4.7 | 1.1 | 3.8 | 1.4 | 0.5 | 1.6 |
| 8 | 6.9 | 1.9 | 8.7 | 10.0 | 3.7 | 11.0 | 1.7 | 1.0 | 3.2 |
| 9 | 6.6 | 2.1 | 4.2 | 6.0 | 0.7 | 6.1 | 5.4 | 3.7 | 8.0 |
| 10 | 5.9 | 1.3 | 7.5 | 6.8 | 0.9 | 6.4 | 5.4 | 3.8 | 4.8 |

**Table 3 Binding free energies of NOR, ROS and NAL.** Binding free energies and corresponding standard deviation (within parentheses) calculated on the most populated cluster of each replica. The cluster population is also reported. Representative clusters reported in boldface are shown in Fig. 7.

| | NOR | | ROS | | NAL | |
|---|---|---|---|---|---|---|
| Replica | $\Delta G$ (SD) kcal.mol$^{-1}$ | Population | $\Delta G$ (SD) kcal.mol$^{-1}$ | Population | $\Delta G$ (SD) kcal.mol$^{-1}$ | Population |
| 1 | **−29.3 (3.8)** | **500** | −24.3 (2.7) | 251 | −20.6 (3.8) | 258 |
| 2 | −25.5 (4.8) | 497 | **−30.5 (2.9)** | **491** | −24.2 (3.1) | 498 |
| 3 | −27.2 (4.6) | 500 | **−29.7 (4.0)** | **297** | −24.7 (3.2) | 500 |
| 4 | **−30.3 (4.1)** | **500** | −24.5 (2.9) | 460 | −24.8 (3.3) | 360 |
| 5 | −28.4 (4.2) | 500 | **−28.4 (2.5)** | **306** | **−26.2 (2.4)** | **500** |
| 6 | −24.7 (3.8) | 500 | −23.6 (3.0) | 500 | **−25.0 (2.3)** | **256** |
| 7 | −28.3 (4.8) | 500 | −23.6 (3.3) | 371 | **−25.5 (3.5)** | **500** |
| 8 | −26.7 (4.3) | 497 | −25.6 (4.8) | 152 | −22.7 (3.2) | 487 |
| 9 | **−32.7 (2.4)** | **287** | −26.8 (4.1) | 500 | −22.1 (2.3) | 289 |
| 10 | −18.0 (4.9) | 429 | −23.3 (3.7) | 496 | −23.0 (3.4) | 500 |
| Average | −27.1 (4.2) | 471 | −26.0 (3.5) | 382 | −23.9 (2.8) | 415 |

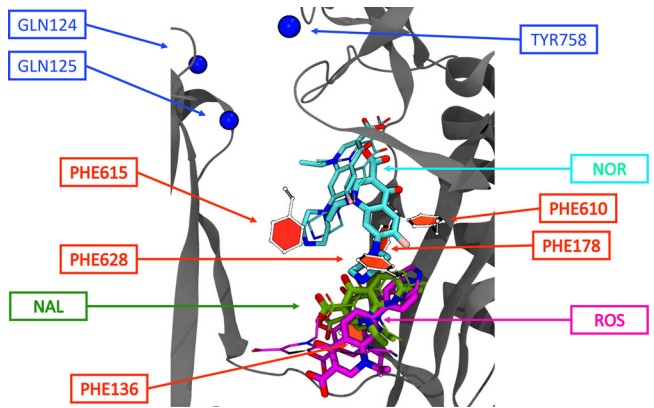

**Fig. 7 Representative structures of the three most stable clusters of NAL, ROS and NOR within the DP$_T$.** The protein is shown in solid gray ribbons (only the PCX and PNX domains are shown for clarity); sidechains of the phenylalanines (PHE) lining the HT are shown using the CPK representation; residues lining the exit gate are highlighted as blue spheres. The NOR, ROS and NAL poses are shown as sticks with carbon atoms coloured cyan, magenta, and green, respectively. The thickest (thinnest) sticks indicate the pose with the lowest (highest) $\Delta G$ value, respectively.

results and with experimental data). Moreover, the analysis of the highest-affinity binding poses within the DP$_T$ confirm that NOR preferentially binds to a different sub-pocket of the DP$_T$ than NAL and ROS (Fig. 7). Namely, NOR loses interactions with the HT and moves upwards to the center of the DP$_T$[15], while both

NAL and ROS establish strong interactions essentially with the whole HT. The two distinct interaction patterns observed for the FQ and the CPTs are clearly traced in Table 4, which reports the most relevant per-residue contributions to the ⊗G for each complex. The most likely complex between NOR and AcrB is stabilized mainly by residues belonging to the upper DP$_T$ (groove)[15], particularly by Q151, F178, S180, I277, and S287. In contrast, F136, Y327, and F628, located in the lower part of this site (cave), are crucial to stabilize CPTs.

## Discussion

In recent years, several studies addressed the transport of antibiotics mediated by bacterial multidrug efflux pumps (for recent reviews see[1,3,4]). However, it remains challenging to establish the molecular bases for the recognition of several antibiotics by a given efflux pump[16–19]. The recent development of theoretical and experimental frameworks to monitor the kinetics of fluoroquinolone accumulation in bacteria expressing different levels of efflux pumps allowed the derivation of chemical structure-efflux efficiency relationships.

In this study, we used resistant clinical strains expressing high efflux activity to identify trends in the intracellular concentrations of fluoroquinolones in correlation with their sensitivity to efflux. We evaluated on whole cell the competition that takes place in the AcrB efflux pump binding sites when two antibiotics are present, by monitoring their intracellular concentration. Among the three non-fluorescent quinolones tested as putative competitors, NAL and ROS are respectively the least and the most effective under the conditions used with ENR, FLE, CIP and NOR. A molar ratio

**Table 4 Per-residue contribution to the binding free energy (kcal.mol⁻¹).**

| Residue | NOR | ROS | NAL |
|---|---|---|---|
| S134 | | −1.0 | |
| S135 | | −1.1 | −1.3 |
| F136 | | −1.3 | −1.7 |
| V139 | | −0.8 | −1.5 |
| Q151 | −1.5 | | |
| S155 | −0.6 | | |
| F178 | −3.3 | | −0.7 |
| G179 | −1.1 | | |
| S180 | −2.2 | | |
| I277 | −2.1 | | |
| I278 | −1.3 | | |
| A279 | −1.1 | | |
| S287 | −2.1 | | |
| Y327 | | −1.1 | −1.3 |
| V571 | | −0.9 | −0.8 |
| M573 | | −0.7 | |
| F610 | −0.7 | | −0.8 |
| V612 | −1.2 | | |
| F615 | −1.2 | | |
| F617 | | −0.8 | |
| F628 | −1.0 | −1.6 | −2.8 |
| L668 | | −1.8 | −0.6 |
| V672 | | −0.9 | |

Values represent averages over three independent MD simulations for each compound (bold in Table 3). Only absolute values larger than kT at room temperature (0.59 kcal.mol⁻¹) are reported.

of five ROS/NOR is necessary to restore an internal concentration of NOR like those observed in the presence of the energy poison. We have previously reported that the use of high antibiotic dose can induce a saturation of efflux capability with the same molecule[8]. This is the first time that a restoration of the internal concentration of an antibiotic substrate of AcrAB-TolC pump is obtained at this level by using a competition approach. Interestingly, the ratios required for the restoration of internal concentration conferred by competitors agree with the SICAR$_{EFF}$ index (Structure Intracellular Concentration Activity Relationship for Efflux) previously reported for CIP, ENR, FLE and NOR, with a highest SICAR$_{EFF}$ for ENR and a lower for NOR[2]. Importantly, restoration is also observed *in cellulo* with individual cell assay, and this threshold of efflux saturation defines the ratio of competitive affinity for the AcrB site illustrating the susceptibility of each quinolone to be expelled across this pump.

In silico studies of the interaction of AcrB with one fluoroquinolone (NOR) and the two quinolones NAL and ROS suggests a possible molecular mechanism of competition. The quinolones bind tightly to the HT within the DP$_T$, similarly to what has been observed for bad substrates or inhibitors of AcrB[2,10,20,21]. According to the binding modes reported in Fig. 7, a small but detectable overlap between the piperazinic ring of NOR and the two quinolones is present. Such a steric clash should clearly impair binding of NOR, resulting in competitive inhibition by ROS and NAL in a way like that suggested in previous studies[16,17]. Alternatively, these two compounds could impair AcrB activity by interfering with the functional conformational cycling of the transporter. Such a mechanism in connection to tight binding to the HT was already proposed to explain the action of some inhibitors, as well as the impaired substrate efflux in the F610A variant of AcrB[22,23].

Our results illustrate the challenge for developing new molecules with reduced affinity for AcrB or appropriate inhibitor targeting AcrB activity. When the site of inhibition is located inside the substrate binding pocket and creates a steric hindrance, the efficacy depends mainly on the corresponding molecular interactions occurring in this pocket[24]. Consequently, some compounds may have no effect or decrease efflux of certain antibiotics such as what is likely the case for the NOR-ROS pair. This is consistent with recent data reporting that an antibiotic and the efflux inhibitor PAßN simultaneously recognize and bind diverse sites in the deep binding pocket of AcrB[23].

Among the tested quinolones, the NOR-ROS pair seems the most interesting one regarding the competition on AcrB sites. Moreover, the restoration level of intracellular antibiotics concentration induced by the NOR-ROS competition was able to generate a notable increase of killing rates.

The efflux competition assay used here can be generalizable and useful for similar studies on bacterial efflux pump systems. Indeed, it combines several key features since the method rely on i) intracellular accumulation measurements in bacterial population or in individual cells, ii) fluorimetry techniques that enable an easy manipulation and the use of an internal standard allowing the comparison between samples, iii) the use of unmodified compounds with anti-bacterial activity. The general design principles of the competition efflux method are described in Fig. 8. With the use of proper antibacterial compounds, strains, and experimental conditions, the method allows to evaluate affinities of AcrB for specific compounds, to measure the contribution of efflux in drug activity, to search for putative competitive substrates, to identify efflux pharmacophores, etc. In the future, we plan to use this method for dissecting the mechanistic of other pumps, such as OqxAB or MexAB in the presence of one or several efflux substrates.

In conclusion, the information gathered by combining experimental methods able to measure the accumulation levels of single antibiotics or combinations of them on whole bacterial cell, individuals or population, and computational protocols with microscopic accuracy will help rationalize the design of new antibacterial strategies.

## Methods

**Bacterial strains and reagents**. EA27 is a multidrug resistant clinical isolate of *Klebsiella aerogenes* exhibiting a mutation in the quinolone resistance-determining region, a porin deficient phenotype and an energy-dependent efflux[9,25]. EA294 is the *acrA*::Kan$^r$ derivative strain of EA289 (not presented in this study), which itself is a kanamycin-sensitive derivative strain of EA27[26]. EA3, EA117, KP 55, ARS100 and ARS108 are multidrug resistant clinical isolates of *Klebsiella aerogenes* (EA3, EA117) *K. pneumoniae* (KP55) and *Escherichia coli* (ARS100, ARS108) exhibiting a porin deficient phenotype and expression of the AcrB efflux pump[27,28].

All strains were cultured in Mueller Hinton II broth (MHII) or Luria Broth (LB) at 37 °C. EA294 was grown in MHII or LB supplemented with kanamycin (50 µg.ml⁻¹). The quinolones ciprofloxacin (CIP), enrofloxacin (ENR), fleroxacin (FLE), nalidixic acid (NAL), norfloxacin (NOR), sparfloxacin (SPA), chloramphenicol and erythromycin were purchased from Sigma-Aldrich. Rosoxacin (ROS) was purchased from LGC Standards Ltd. The quinolones were dissolved in Milli-Q water supplemented with 0.1 to 0.4% (v/v) NaOH, chloramphenicol and erythromycin were dissolved in 50% ethanol. Physicochemical parameters of the studied quinolones are listed in Supplementary Fig. 1.

**Determination of minimum inhibitory concentrations (MICs)**. MICs were determined by the standard 2-fold microdilution method in 96-well microplates according to the guidelines of the clinical & laboratory standards institute (CLSI, http://clsi.org/). Cultures were grown in MHII and approximately 2 × 10⁵ cells were inoculated to each well. The MIC results were read by the unaided eye after incubation at 37 °C for 18 h as the lowest concentration of antimicrobial agent that completely inhibits growth of the organism. Experiments were carried out at least in triplicate.

**Resazurin-reduction-based bacterial viability assay**. The resazurin-reduction-based bacterial viability assay was used to evaluate the effect of the studied quinolones on the cell's metabolic activity. When resazurin (non-fluorescent, blue-colored compound) penetrates bacterial cell, it is reduced by metabolically active

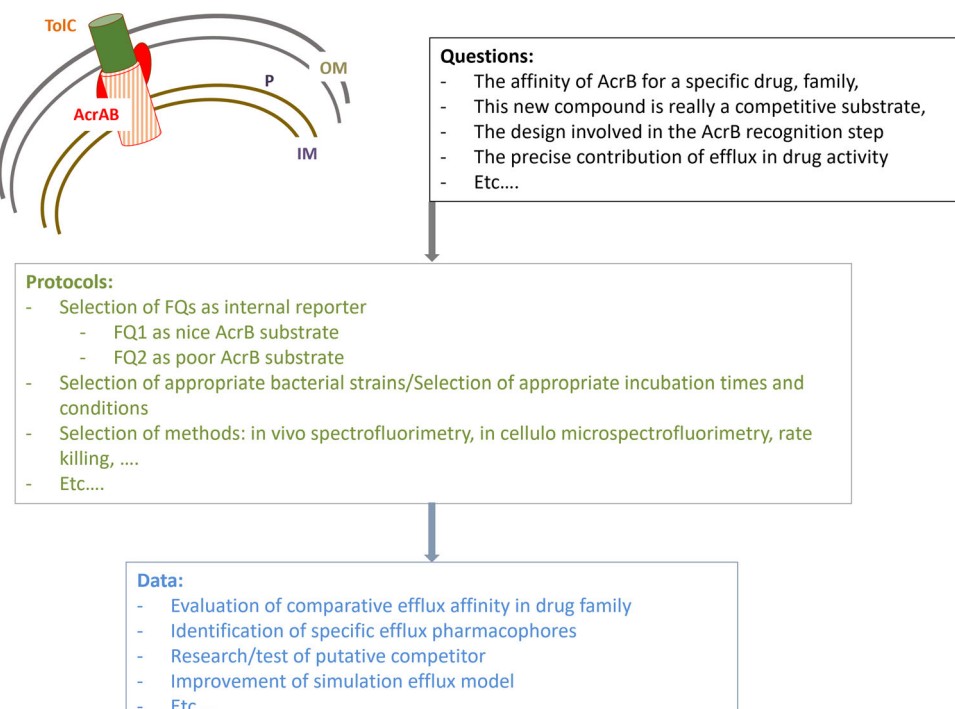

**Fig. 8 General design principles of the competition efflux method.** The efflux competition assay used here can be extended to other drugs using appropriate protocols to investigate several questions about bacterial efflux pump systems. The figure shows examples of these questions and protocols, as well as the type of data that can be obtained from this assay.

bacterial cells to resorufin (fluorescent, pink-colored product); which results in a color change that can be determined by measurement of the fluorescent signal of resorufin. Cultures were grown at 37 °C in MHII to mid-exponential phase, then diluted to around $2 \times 10^7$ cells.ml$^{-1}$ in fresh MHII. In 96-well clear bottom black microplates, 170 µl of the bacterial suspension was mixed with 20 µl of the resazurin dye (Cell-Titer Blue® Cell-Titer Viability Reagent, Promega) and 10 µl of individual quinolone at various concentrations. Control wells also contained cells with resazurin but no quinolone, and resazurin with quinolone without cells. Fluorescent signals of resorufin were measured at 37 °C every 10 minutes for 5 hours with a TECAN Infinite Pro M200 spectrofluorometer ($\lambda$ex = 530 nm and $\lambda$em = 590 nm). The metabolic inhibition rate of each strain exposed to each quinolone was calculated from the difference of relative fluorescence units in the presence as compared to in the absence of quinolone, at the first time point of the fluorescence plateau measured in the control condition (plateau corresponding to 100% fluorescence / 100% viability). All experiments were performed at least 3 times.

**Cell killing rates.** Bacterial suspensions were sampled during accumulation assays for CFU and subsequently cell killing rate determination. Suspensions were consecutively diluted in fresh LB and spread in Petri dishes incubated overnight at 37 °C. CFU were then used to determinate the killing rates during incubation. Killing rates determination was performed in triplicate at three independent times.

**FQ accumulation assay.** Bacteria were grown at 37 °C in LB to mid-exponential phase. The bacterial suspension was centrifuged at 6,000 $g$ for 15 min at 20 °C and concentrated 10 times by re-suspension of the pellets in 1/10 of the initial volume in 50 mM sodium phosphate buffer at pH 7 supplemented with 5 mM MgCl$_2$ to obtain a density of 6×10$^9$ CFU (colony forming units) per ml[6]. In glass culture tubes, the bacterial suspension was incubated at 37 °C in the absence or in the presence of each FQ at 8 µg.ml$^{-1}$ (corresponding to concentrations of 22 to 25 µM depending on the FQ) (Figs. 1, 3 and Supplementary Figs. 3, 4, 5, 6) or at 16 µM (corresponding to 5 µg.ml$^{-1}$ of NOR) (Fig. 4), in the absence or in the presence of CCCP at 20 µM, in the absence or in the presence of CPTs, chloramphenicol or erythromycin (Supplementary Fig. 4) at various concentrations. Concentrations of CPTs, chloramphenicol and erythromycin added during incubation are relative to the co-incubated FQ concentration in molar equivalent. The bacterial suspensions incubated in the absence of FQs, in the absence or in the presence of CCCP, in the absence or in the presence of CPTs, chloramphenicol, erythromycin were used as controls. Suspensions of the incubation mixture sampled at 20 minutes (Figs. 1, 3 and Supplementary Figs. 3, 4, 6), 5 and 20 minutes (Supplementary Fig. 5) or 30 and 60 minutes (Fig. 4) were loaded on sodium phosphate buffer and centrifuged at 9000 $g$ for 5 min at 4 °C to collect the washed bacteria[6]. Accumulation assays for

each tested condition were performed in triplicate in at least three independent times (Figs. 1 and 4).

**Spectrofluorimetry.** Consecutively to the accumulation assay, the pellets of washed bacteria were lysed with 500 µl of 0.1 M Glycin-HCl pH 3 overnight at room temperature. After a centrifugation for 15 min at 9000 $g$ at 4 °C, 50 µl of each sampled lysate were placed in black 96-well microplates and emission spectra were measured with a TECAN Infinite Pro M200 spectrofluorometer[6]. Excitation/emission range wavelengths used for detection of FQs and tryptophan fluorescence signals were 275/300-500 nm for CIP, ENR and NOR, and 290/310-500 nm for FLE. The fluorescence signals of FQs from each sample were corrected by the tryptophan fluorescence to obtain a fluorescence signal of FQ per bacterial cell[6]. In parallel, increasing concentrations of FQs were mixed with lysates of bacteria resulting from incubations without FQ, in the absence or in the presence of CCCP, in the absence or in the presence of CPTs, chloramphenicol or erythromycin, in order to determine the calibration curves used to calculate the number of FQ molecules accumulated per cell[6].

**Deep-ultraviolet microspectrofluorimetry.** To detect the NOR fluorescence from single bacteria background, the pellets of washed bacteria obtained from accumulation assays were re-suspended in 40 µl of sodium phosphate buffer. 0.5 µl of resuspended pellets were deposited between two quartz coverslips and analyzed by deep-ultraviolet fluorescence imaging at DISCO Beamline (Synchrotron SOLEIL)[29]. Bacterial cells were first located in brightfield before excitation in deep-ultraviolet under a microscope (Zeiss Axio Observer Z-1). Emission was collected through a Zeiss ultrafluar objective at 100 x with glycerine immersion. The NOR fluorescence intensities were recorded after excitation using a dichroic mirror at 300 nm (OMEGA Optical, Inc., USA) through appropriate emission bandpass filters (OMEGA Optical, Inc., USA; SEMROCK, USA). Excitation/emission filters to detect the NOR signal were 275/420-480 nm. For the tryptophan detection, fluorescence was selected through an emission bandpass filter at 327-353 nm (SEMROCK). Intensities in tryptophan filter were collected immediately after the drug fluorescence signal was acquired. Fluorescence images were recorded by a back-illuminated ultraviolet electron-multiplying charge-coupled device (Princeton PIXIS 1024 BUV). The whole setup (microscope, stages, filters, camera) was controlled by Micro-Manager[30].

The images were analyzed with a set of python scripts in a jupyter notebook (https://gitlab.synchrotron-soleil.fr/disco-beamline/bacteria-drug-uv-analysis). The NOR fluorescence intensities were normalized by intensities measured in tryptophan filter and by the mean value of the ratio of the fluorescence intensities collected in NOR filter to the ones collected in the tryptophan filter from the corresponding control samples (without NOR)[6]. For each condition, at least 6 different locations with minimum 20 bacteria per field of view were recorded. The experiment was independently repeated two times.

**Molecular simulations of quinolones interacting with AcrB.** To investigate the interaction between quinolones and AcrB we combined molecular docking and all-atom MD simulations. Molecular docking calculations were performed using AutoDock Vina[31] which implements a stochastic global optimization approach. Following previous studies[2,10], the program was used with default settings but for the exhaustiveness parameter which was set to 1024 (default 8). Ligand protonation states were calculated using Marvin Calculator Plugin (Marvin Calculator Plugin, version 19.14.0 2011 provided by Chemaxon (www.chemaxon.com)). Compounds were considered flexible during docking (number of rotatable bonds in the range 2 to 5). Protein and ligand input files were prepared in PDBQT file format with AutoDock Tools[32]. For *K. aerogenes* no crystal structures of AcrB are available, thus, given the high homology degree of AcrB in *E. coli* and *K. aerogenes* and the conservation of the region of interest (see Supplementary Fig. 10), we used an ensemble of 18 X-ray conformations of AcrB to account of protein flexibility. We adopted a cubic box of dimensions 30 Å x 30 Å x 30 Å, centred in the centre of mass of the $DP_T$ (Fig. 5a). The top 10 docking poses were retained for each run and the top scoring ones were selected for further analysis (from a total number of 180 poses per compound). Molecular graphics were rendered using VMD (visual molecular dynamics)[33].

The initial coordinates of the AcrB-molecule complexes were taken from selected docking poses. Like in our previous study[2], we employed a reduced model of the AcrB protein not containing the transmembrane domain, that has been validated in previous studies[15,20]. Each system was solvated in a box containing TIP3P water molecules[34], and an adequate number of $K+$ counterions, in order to neutralize the negative net charge of the system. An osmolarity of 0.15 M was reached by adding an appropriate number of $K+/Cl-$. The ff14SB version of the AMBER force field[35] and the General Amber Force-Field parameters[36] were adopted for AcrB and for each molecule, respectively. The systems were then minimized with a combination of steepest descent and conjugate gradient methods gradually releasing positional restraints applied. The protocol adopted for MD simulations is the same reported in Ref. [10].

Root mean square deviations (RMSDs) were calculated using the *cpptraj* module of AMBERTools and VMD. A cluster analysis using *cpptraj* identified the most populated configurations sampled during the simulation with a fixed clustering radius of 3.5 Å. Only the most populated cluster were further considered for evaluating the free energies of binding using the molecular mechanics generalized Born surface area[37] approach. The reported binding free energies refer to the unrestrained 50 ns-long production phase, while RMSDs refer to the whole 100 ns trajectory.

**Statistics and Reproducibility.** The statistical analyses of experimental data were performed using the computing environment R (R Core Team (2020)). ANOVAs with Tukey's post-hoc tests were used to determine differences of FQs accumulation shown in Fig. 1 (At least three independent assays carried out in triplicate, $n \geq 3$) and differences of killing rates shown in Fig. 4 (Three independent assays, $n = 3$). P-values between 0.01 to 0.05: significant (*), P-values between 0.001 to 0.01: very significant (**), P-values < 0.001: extremely significant (***). The data normality and homoscedasticity were checked by the respective Shapiro-Wilk and Fligner-Killeen tests. Kruskal-Wallis with Dunn tests were used to determine differences of NOR accumulation shown in Fig. 3 (Two independent assays, 6 microscopy fields, each containing about 20 bacteria, n ranging from 200 to 423) and 4. Figure 2 shows means of FQs accumulation % (±SD) obtained with each CPTs calculated from data plotted in Fig. 1 considering the 0% and 100% as the FQ accumulation measured in the absence of CPT without (0%) and with CCCP (100%). Trend polynomial curves of degree 2 were plotted (Pearson correlation ≥ 0.96) and were used to calculate the CCIA indexes. Figure 6 reports the average binding affinities for each ligand as obtained through molecular docking performed using an ensemble of 18 X-ray conformations of AcrB. The corresponding standard deviations are reported ($n = 180$).

Three independent assays were performed to obtain MIC data (Table 1, $n = 3$). Table 3 reports the binding free energies and the corresponding standard deviation calculated on the most populated cluster of each replica. The number of frames ($n$, ranging from 152 to 500) are also reported.

Binding free energies shown in Table 4 were from three independent MD simulations ($n = 3$). Three independent assays were carried out to obtain data of Supplementary Fig. 4. One or two independent assays were performed in triplicate to obtain the data from Supplementary Figs. 3, 5 and 6.

Independent assays: biologically and temporally independent samples; triplicate: technical replicates.

**Reporting summary**. Further information on experimental design is available in the Nature Research Reporting Summary linked to this paper.

## Data availability

Source data are available (Supplementary Data 1). All other data supporting this study are available within the article and its Supplementary Information or are available from the corresponding author on reasonable request.

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

## Acknowledgements

We thank R.A. Stavenger, M. Winterhalter, A. Davin-Regli, JM. Bolla and M. Masi for their fruitful discussions; B. Pineau and V. Rouam for their assistance during the assays. This work was supported by Aix-Marseille Univ., INSERM, Service de Santé des Armées, by Soleil program (projects #20181559, #20190254, #20191701), and by National Institute of Allergy and Infectious Diseases/NIH grant no. R01AI136799. In Memoriam to Leonard Amaral, who died in 2020, an active member of the COST European program ATENS focused on antibiotic efflux. He has strongly contributed to the efflux studies carried out in several European laboratories and stimulated lot of research in the domain during recent decades.

## Author contributions

J.V., H.C., F.O., J.P., G.M., A.V.V.: investigation, data curation. M.R., P.R., J-M.P.: resources, supervision. All the co-authors: writing, review, and final editing.

## Competing interests

The authors declare no competing interests.
