## [Peer Review File · Communications Biology]

Reviewers' comments:

Reviewer #1 (Remarks to the Author):

A study by Vergali, Chauvet, Oliva, and colleagues provides an experimental and computational framework that addresses antibiotic transport by efflux pumps. They focus on a specific class of clinically-relevant antibiotics: quinolones. Used methods allow authors to study efflux capabilities in populations of bacteria and single cells. To corroborate their findings, researchers employ computational methods to pinpoint the residues of AcrB that affect the interaction of the pump with chosen quinolones.

The manuscript describes the molecular foundation that underlies the recognition of antibiotics by efflux pumps. Using resistant clinical isolates, authors evaluate the competition of two antibiotics within the AcrB efflux pump. Authors find that non-fluorescent quinolones compete with fluorescent quinolones more or less effectively. Specifically, NAL is less effective than ROS.

Supporting computations identify potential structural causes for observed competition. Further, the results demonstrate how designing new antibiotics that have a lower affinity for AcrB is difficult, since steric hinderance and molecular interactions occurring within binding pocket must be accounted for.

To this reviewer's knowledge, such in vivo competition of antibiotics within the AcrB pump has not been described before.

Amidst the looming antibiotic crisis, mechanistic understanding of resistance determinants is of importance. Both conceptual and methodological work (such as this one) are crucial for finding effective ways of battling the spread. The manuscript presents an intriguing combination of experiments and computational tools that appear general enough to be of interest for researchers looking into antibiotic transport phenomena. As such, this work has the potential to influence how researchers approach studying phenomena similar to the ones described in the manuscript. Overall, the claims I could assess within my expertise seem to be technically sound, sufficiently well described, and are of interest. I cannot comment on the details of the computational part.

Authors use relatively complex prose to describe their findings, based on often convoluted and long sentences. I understand that authors resort to abbreviations to shorten the text. I would advise that in the Discussion authors refrain from using abbreviations and spell out the terms: this makes the reader reflect on rather than decipher the text.

The authors' style of reporting is very objective, to the point it sometimes lacks the "enthusiasm." However, this is a stylistic choice and should be left to the editors and authors. I do not find the claims overblown. Combined with the authors' previous work (Vergalli and Atzori et al., *Comm Biol* 2020), the level of detail in the text is sufficient for a researcher with significant prior experience to reproduce the described research but it might not be accessible to novices in the field.

Specific comments

-The deposition of software on a public server for broader use is commendable. However, I would ask the authors to translate the software comments and description text.

-I think the discussion should be made more general. I find the method to be generalizable to similar studies. If this is the case, I think the Discussion would benefit from describing the general "design principles" that would guide the other researchers when adapting this framework to other problems.

Minor

L345-347: Sentence needs a clarification (e.g., by inserting "which") "EA294 is ... , WHICH itself is"
L354: solubilized -> dissolved; solubilization refers to making a compound more soluble.

L362-363: clarify what "results were read" means. Does it refer to measuring optical density when determining MIC? If so, please add the wavelengths.

L451: localizations->locations?

Reviewer #2 (Remarks to the Author):

This is an interesting study that offers insight into the effect of structurally related substrate antibiotics on AcrAB efflux pump and their accumulation within the bacteria. The study has the potential to be useful for the researchers working on antimicrobial resistance and developing new tools to study efflux pump systems in bacteria. The authors should address the following comments in the revised version of the manuscript.

-

- The title should be amended as the study doesn't provide information about the affinity of antibiotics for AcrAB. It provides some limited information about the substrate – pump relationship for structurally related quinolone class of antibiotics. It is not possible to extend this to structurally unrelated antibiotic classes without further study. It should state "The quinolone transport by bacterial efflux pump: an original competitive approach for dissecting AcrB affinities for fluoroquinolones".

Past studies involving AcrAB efflux system in E coli (Antimicrobial agents and chemotherapy, 54(9), 3770-3775.) showed limited competition between different set of substrate antibiotics which suggest a study limited to fluoroquinolones is unlikely to inform competition and AcrB affinity for other classes of antibiotics. The authors should at least include another class of AcrB substrate to validate the findings of the current study. They should repeat the study by replacing the non-fluorescent fluoroquinolones with another AcrB substrate like chloramphenicol, rifampin, macrolides or novobiocin.

The authors have published extensively using the EA27 model, and they mention in their discussion that "we used resistant clinical strains expressing significant efflux activity to identify trends in the intracellular concentrations of fluoroquinolones in correlation with their sensitivity to efflux.". In reality they have only used one resistant clinical strain. There are a large number of clinical strains available that over-expresses AcrAB. It would be useful to widen the study to few more clinical strains that overexpress AcrAB to check the validity of findings reported in this study,

The impact of the study on the study of efflux pump-substrate antibiotic has been over-emphasised in this study. Co-administration of efflux pump substrates often shows synergy and many studies have reported interaction between different or same classes of antibiotics or efflux pump inhibitors through checkerboard and other competition assays. It has been widely reported that AcrAB has a large number of structurally diverse substrates and the capacity of this efflux system is very high (Antimicrob. Agents Chemother. 51:923-929.). Is there any notable differences in the capacity of AcrAB efflux pump system in MDR E coli vs resistant Klebsiella aerogenes EA27?

The cooperative cell killing rates due to increased accumulation of FQs should be extended to other FQ and CPT combinations and over a longer time point. The authors only study NOR-ROS combinations. It is not clear why the author selected these two FQ antibiotics over others. It is unlikely the cell killing effect can be observed within 30 minutes and 60 minutes of co-administration. It would be helpful to observe the effect of accumulation on cell killing in both AcrAB overexpressing resistant strains, AcrAB null strains and wild type strains.

Minor point: the authors uses in vivo with reference to interaction of compounds with efflux pumps in bacteria. The nature of the study conducted in the manuscript should be considered in vitro and should be changed accordingly throughout the manuscript.

Reviewer #3 (Remarks to the Author):

Vergalli et al. present a detailed study of the quinolone efflux capacities of the AcrB multidrug efflux pump in *Klebsiella aerogenes*, and explore competitive inhibition of the pump as a way to enhance antibacterial activity. The experimental approach is innovative and the results are interesting and potentially valuable to the field. However, the study could be better motivated and the story more coherent. A more clearly structured, well-motivated, and to-the-point text would strongly improve the readability of the manuscript and clarify the significance of the results. This comment concerns the entire text, but special attention should be given to the abstract, where the key findings could be stated more clearly and specifically.

In addition, I have some comments regarding the methodology and presentation of the results.

Major comments:

- 1) Lines 119-128: What is the relevance of measuring the effect of CPTs, but not FQs, on metabolic activity? Why not measure bactericidal activity of all compounds using CFU-based killing assays (i.e. using a similar protocol as in Figure 4)?
- 2) As CCCP might have additional, unintended effects on bacterial cells, it would be useful to have EA249 as an additional no-efflux control for the results presented in Figure 1-4.
- 3) The presence of the QRDR mutation in EA27 might mask the potential of competitive inhibition to promote bacterial killing, as FQs might have reduced bactericidal effects despite increased intracellular concentration. For this reason, it would be useful to perform the killing assays with strains lacking the QRDR mutation, with and without AcrB.
- 4) It is not entirely clear to me whether the presented results suggest the use of antibiotic combination therapies to enhance antibacterial activity. As the authors suggest that the effect of competition is a result of efflux pump saturation, can a similar effect be attained using monotherapy with higher FQ concentrations (similar concentration as the total concentration in the combinations)? Elaborating on the benefits of using CPTs could highlight the significance of the results.
- 5) Related to the comment above, it would be useful to discuss where the combined concentrations (FQ+CPT) are situated relative to the total concentration needed for AcrB saturation. Do the observed competition effects differ when moving closer or further from this plateau concentration? (In this context, please also clarify lines 303-305.)

Minor comments:

- 6) The subtitles in the Results section could be made more informative.
- 7) Lines 106-108: When looking at Table 1, the MICs for EA27 seem quite a bit more than 9 to 10-fold increased compared to AG100.
- 8) Please always use the same units for antibiotic concentrations (either μM or $\mu\text{g/ml}$).
- 9) The *E. coli* AG100 strain was used for MIC testing, but these results do not seem very relevant for the discussion and no further experiments were done on this strain, so it seems like these data can be left out.
- 10) Lines 175-219: Please motivate why you only focus on NOR in this section.
- 11) Lines 491-493: Please describe in more detail the model fitted to the data in Fig. 2A.
- 12) Fig. 3: Please show significant differences, similar to Fig. 1.
- 13) Lines 214-216: By comparing NOR+ROS only to NOR and ROS, it seems hard to disentangle the direct effect ROS on bacterial killing and its indirect effect through competition. I think it would be useful to involve the NOR+CCCP result in the discussion, as this is supposed to indicate the maximal effect on killing that can be attained by competitive pump inhibition alone.

Reviewers' comments and responses :

Reviewer #1 (Remarks to the Author):

A study by Vergali, Chauvet, Oliva, and colleagues provides an experimental and computational framework that addresses antibiotic transport by efflux pumps. They focus on a specific class of clinically-relevant antibiotics: quinolones. Used methods allow authors to study efflux capabilities in populations of bacteria and single cells. To corroborate their findings, researchers employ computational methods to pinpoint the residues of AcrB that affect the interaction of the pump with chosen quinolones.

The manuscript describes the molecular foundation that underlies the recognition of antibiotics by efflux pumps. Using resistant clinical isolates, authors evaluate the competition of two antibiotics within the AcrB efflux pump. Authors find that non-fluorescent quinolones compete with fluorescent quinolones more or less effectively. Specifically, NAL is less effective than ROS.

Supporting computations identify potential structural causes for observed competition. Further, the results demonstrate how designing new antibiotics that have a lower affinity for AcrB is difficult, since steric hindrance and molecular interactions occurring within binding pocket must be accounted for.

To this reviewer's knowledge, such in vivo competition of antibiotics within the AcrB pump has not been described before.

Amidst the looming antibiotic crisis, mechanistic understanding of resistance determinants is of importance. Both conceptual and methodological work (such as this one) are crucial for finding effective ways of battling the spread. The manuscript presents an intriguing combination of experiments and computational tools that appear general enough to be of interest for researchers looking into antibiotic transport phenomena. As such, this work has the potential to influence how researchers approach studying phenomena similar to the ones described in the manuscript. Overall, the claims I could assess within my expertise seem to be technically sound, sufficiently well described, and are of interest. I cannot comment on the details of the computational part.

We thank the Reviewer for the positive feedback on our work.

Authors use relatively complex prose to describe their findings, based on often convoluted and long sentences. I understand that authors resort to abbreviations to shorten the text. I would advise that in the Discussion authors refrain from using abbreviations and spell out the terms: this makes the reader reflect on rather than decipher the text.

We followed the appropriate advice of the Reviewer; we have, as far as possible, reduced the use of abbreviations in the discussion section. Additionally, after the list of Keywords we have introduced an alphabetically ordered list of all the abbreviations used in the manuscript.

The authors' style of reporting is very objective, to the point it sometimes lacks the “enthusiasm.” However, this is a stylistic choice and should be left to the editors and authors. I do not find the claims overblown. Combined with the authors' previous work (Vergalli and Atzori et al., Comm Biol 2020), the level of detail in the text is sufficient for a researcher with significant prior experience to reproduce the described research but it might not be accessible to novices in the field.

We thank the Reviewer for the comment. We are convinced that our work contains very interesting results and, accepting Reviewer's suggestion, we have inserted sentences summarizing with more emphasis our achievements and the perspectives opened by our work. We also revised some sentences of the manuscript in order to make it more accessible to novices in the field.

Specific comments

The deposition of software on a public server for broader use is commendable. However, I would ask the authors to translate the software comments and description text.

We have officially released the analysis toolbox already. It is still in active development, and not yet ready for broader use. One of its actual limitations lies in its strong dependence on the DISCO Beamline data acquisition workflow. As suggested by the Reviewer, we have translated comments of each function as far as possible.

I think the discussion should be made more general. I find the method to be generalizable to similar studies. If this is the case, I think the Discussion would benefit from describing the general “design principles” that would guide the other researchers when adapting this framework to other problems.

We fully agree with the suggestion of the Reviewer. We have added a paragraph in the Discussion section and a figure, namely Figure 8, which describes the general principles of design of the competition method developed in this study.

Minor

L345-347: Sentence needs a clarification (e.g., by inserting “which”) “EA294 is ... , WHICH itself is” L354: solubilized -> dissolved; solubilization refers to making a compound more soluble.

L362-363: clarify what “results were read” means. Does it refer to measuring optical density when determining MIC? If so, please add the wavelengths.

L451: localizations->locations?

We thank the Reviewer for the comments, the corresponding sentences have been corrected.

Reviewer #2 (Remarks to the Author):

This is an interesting study that offers insight into the effect of structurally related substrate antibiotics on AcrAB efflux pump and their accumulation within the bacteria.

The study has the potential to be useful for the researchers working on antimicrobial resistance and developing new tools to study efflux pump systems in bacteria. The authors should address the following comments in the revised version of the manuscript.

We thank the Reviewer for this advice, we have added a paragraph and the new Figure 8 in the Discussion section to highlight the potential offered by the competition method developed in this study.

The title should be amended as the study doesn't provide information about the affinity of antibiotics for AcrAB. It provides some limited information about the substrate – pump relationship for structurally related quinolone class of antibiotics. It is not possible to extend this to structurally unrelated antibiotic classes without further study. It should state “The quinolone transport by bacterial efflux pump: an original competitive approach for dissecting AcrB affinities for fluoroquinolones”.

We agree with the Reviewer, we have changed the title and replaced ‘antibiotics’ with ‘fluoroquinolones’.

Past studies involving AcrAB efflux system in E coli (Antimicrobial agents and chemotherapy, 54(9), 3770-3775.) showed limited competition between different set of substrate antibiotics which suggest a study limited to fluoroquinolones is unlikely to inform competition and AcrB affinity for other classes of antibiotics. The authors should at least include another class of AcrB substrate to validate the findings of the current study. They should repeat the study by replacing the non-fluorescent fluoroquinolones with another AcrB substrate like chloramphenicol, rifampin, macrolides or novobiocin.

We thank the Reviewer for this interesting remark. We reproduced the experiment with chloramphenicol and erythromycin (macrolide) (Figure S4). The results obtained with these competitors show an effect close to the one obtained with nalidixic acid, validating the stronger specificity of the competition taking place in AcrB for the NOR-ROS pair as compared to NOR-NAL.

The authors have published extensively using the EA27 model, and they mention in their discussion that “we used resistant clinical strains expressing significant efflux activity to identify trends in the intracellular concentrations of fluoroquinolones in correlation with their sensitivity to efflux.”. In reality they have only used one resistant clinical strain. There are a large number of clinical strains available that over-expresses AcrAB. It would be useful to widen the study to few more clinical strains that overexpress AcrAB to check the validity of findings reported in this study,

Following the suggestion, additional competition efflux assays in various MDR clinical isolates of *K. aerogenes*, *K. pneumoniae* and *E. coli* have been carried out (see Figure S5). Although some slight differences were observed depending on the specific resistant strain

considered, the results obtained validate the extensive study performed with the EA27 model.

The impact of the study on the study of efflux pump-substrate antibiotic has been over-emphasised in this study. Co-administration of efflux pump substrates often shows synergy and many studies have reported interaction between different or same classes of antibiotics or efflux pump inhibitors through checkerboard and other competition assays. It has been widely reported that AcrAB has a large number of structurally diverse substrates and the capacity of this efflux system is very high (Antimicrob. Agents Chemother.51:923-929.). Is there any notable differences in the capacity of AcrAB efflux pump system in MDR *E. coli* vs resistant *Klebsiella aerogenes* EA27?

We agree with the Reviewer that other competition assays have been previously performed. Concerning the checkerboard assay, it only measures a total concentration needed to inhibit growth after 18 hours and does not measure the intracellular concentration in the early step of co-incubation, when the efflux mechanism can be investigated and play the most important role.

In our study, the efflux competition assays carried out are based on i) intracellular accumulation measurements, ii) fluorimetry methods that allow an easy manipulation and the use of an internal standard allowing the comparison of results between samples, iii) the use of unmodified compounds which display anti-bacterial activity. In the Conclusions section we emphasized the above features of the presently developed efflux competition assay.

Concerning the possible difference between MDR *E. coli* and *K. aerogenes*, we have performed additional competition efflux assays in two MDR clinical isolates of *E. coli* (ARS100 and ARS108, Figure S5). The results obtained were consistent with those obtained with the *K. aerogenes* EA27 strain.

The cooperative cell killing rates due to increased accumulation of FQs should be extended to other FQ and CPT combinations and over a longer time point. The authors only study NOR-ROS combinations. It is not clear why the author selected these two FQ antibiotics over others. It is unlikely the cell killing effect can be observed within 30 minutes and 60 minutes of co-administration. It would be helpful to observe the effect of accumulation on cell killing in both AcrAB overexpressing resistant strains, AcrAB null strains and wild type strains.

For rationally approaching the hypothesized existing kinetic correlation between internal accumulation and antibiotic activity, it is necessary to measure the two parameters at same time. Taking into account this constraint and for technical/methodological reasons we have selected two time, 30mins and 60mins. During the determination of SICAR index, we have previously demonstrated that this period correspond to accumulation plateau (see - Nat. Protoc. 2018 10.1038/s41579-019-0294-2; - Sci Rep. 2017 Aug 29;7(1):9821. doi: 10.1038/s41598-017-08775-4; - Sci Rep. 2015 Dec 11;5:17968. doi: 10.1038/srep17968.). In addition, with the data obtained during accumulation assay, it is clear the selected pair for the killing rate is the pair giving the more interesting data in accumulation, e.g. NOR-ROS. Regarding other bacterial strains, the accumulation has been performed and now included in supData.

Minor point: the authors use *in vivo* with reference to interaction of compounds with efflux pumps in bacteria. The nature of the study conducted in the manuscript should be considered *in vitro* and should be changed accordingly throughout the manuscript.

We agree with the Reviewer that the use of the terms *in vivo*/*in vitro* in the manuscript could generate some confusion. In order to clarify this point, the accumulation assay using incubation + cell lysates has been denoted “*in vitro* assay”, the accumulation assay performed on intact cells and direct fluorimetry has been denoted as “*in cellulo* assay”. This latter is based on microspectrofluorimetry which is useful to see the fluorescence in intact individual cells (Nat. Protoc. 2018 10.1038/s41579-019-0294-2). The same issue has been clarified in the manuscript.

Reviewer #3 (Remarks to the Author):

Vergalli et al. present a detailed study of the quinolone efflux capacities of the AcrB multidrug efflux pump in *Klebsiella aerogenes*, and explore competitive inhibition of the pump as a way to enhance antibacterial activity. The experimental approach is innovative and the results are interesting and potentially valuable to the field.

However, the study could be better motivated and the story more coherent. A more clearly structured, well-motivated, and to-the-point text would strongly improve the readability of the manuscript and clarify the significance of the results. This comment concerns the entire text, but special attention should be given to the abstract, where the key findings could be stated more clearly and specifically.

We thank the Reviewer for the comments. The text has been thoroughly modified to improve the readability of our study.

In addition, I have some comments regarding the methodology and presentation of the results.

Major comments:

1) Lines 119-128: What is the relevance of measuring the effect of CPTs, but not FQs, on metabolic activity? Why not measure bactericidal activity of all compounds using CFU-based killing assays (i.e. using a similar protocol as in Figure 4)?

Regarding the selection of the FQ antibiotics used during this killing assay we have selected the molecule that exhibit the more efficient effect on accumulation.

Concerning the metabolic activity measurements, they give us additional information about the compounds activities since it measures an earlier effect of compounds incubated with bacteria (3 to 5 hours compared to 18 h with MIC determination assays). We initially chose to present only the results obtained with the CPTs, which have shown high MICs (NAL and ROS), to highlight discrepancies between MIC/resazurin assays, particularly relevant for the ROS efflux susceptibilities measured with both methods. Following the suggestion of the Reviewer, we added the measure of FQs effect on metabolic activities.

2) As CCCP might have additional, unintended effects on bacterial cells, it would be useful to have EA249 as an additional no-efflux control for the results presented in Figure 1-4.

We followed the Reviewer's suggestion and reproduced the competition experiments on EA294 (no-efflux strain) (Figure S3). The results obtained validate the effects of CPTs on the accumulation of FQs in the EA27 (overexpression of AcrAB) since NOR and ENR accumulation remained high and similar in the no-efflux strain, with and without CPTs and at different concentration of the CPT.

3) The presence of the QRDR mutation in EA27 might mask the potential of competitive inhibition to promote bacterial killing, as FQs might have reduced bactericidal effects despite increased intracellular concentration. For this reason, it would be useful to perform the

killing assays with strains lacking the QRDR mutation, with and without AcrB.

For the determination of cell killing, it is mandatory to consider the various concentrations used for putative competitors that can kill the studied bacteria cells (For instance, the high chloramphenicol concentrations have induced a high cell killing). In order to bypass this key point, we have used bacterial strain expressing mutations in the target (QRDR in gyrase/topoisomerase in EA27).

4) It is not entirely clear to me whether the presented results suggest the use of antibiotic combination therapies to enhance antibacterial activity. As the authors suggest that the effect of competition is a result of efflux pump saturation, can a similar effect be attained using monotherapy with higher FQ concentrations (similar concentration as the total concentration in the combinations)? Elaborating on the benefits of using CPTs could highlight the significance of the results.

We do not suggest “the use of antibiotic combination therapies to enhance antibacterial activity”. We don’t claim to place this study at a clinical level by providing new therapy protocols. The objective of the study is to evaluate the respective affinities between quinolones for efflux pumps and demonstrate the relevance of the competitive method developed here. A new figure (Fig8) has been added to summarize the impact of this work (see also Referee1 ‘s comment)

5) Related to the comment above, it would be useful to discuss where the combined concentrations (FQ+CPT) are situated relative to the total concentration needed for AcrB saturation. Do the observed competition effects differ when moving closer or further from this plateau concentration? (In this context, please also clarify lines 303-305.)

We thank the Reviewer for the comment and we agree that this point is very interesting. Unfortunately, due to technical/methodological constraints, it is not possible today to obtain precise measurements before and after reaching the plateau (time necessary to record the accumulation signal of studied molecules and internal control, time necessary to record several microscopy fields with sufficient cells...) (Nat. Protoc. 2018; 10.1038/s41579-019-0294-2). Concerning specific points the text has been improved.

Minor comments:

6) The subtitles in the Results section could be made more informative.

Subtitles have been changed to be more informative.

7) Lines 106-108: When looking at Table 1, the MICs for EA27 seem quite a bit more than 9 to 10-fold increased compared to AG100.

We thank the Reviewer for the remark. We apologize for this mistake, and, following the suggestion at point 9) we have deleted these data.

8) Please always use the same units for antibiotic concentrations (either μM or $\mu\text{g/ml}$).
The correspondences $\mu\text{g/ml}$ - μM have been added in the 'Methods' section.

9) The *E. coli* AG100 strain was used for MIC testing, but these results do not seem very relevant for the discussion and no further experiments were done on this strain, so it seems like these data can be left out.

We agree, we have deleted these data.

10) Lines 175-219: Please motivate why you only focus on NOR in this section.

For the corresponding paragraphs we have selected NOR since the NOR-ROS pair exhibits the more efficient effect on accumulation. We have added additional data and modified the text to highlight why we focused on NOR in this section.

11) Lines 491-493: Please describe in more detail the model fitted to the data in Fig. 2A.

The trend curves plotted in Figure 2A are degree 2 polynomials. We have added this information in the Figure 2 legend.

12) Fig. 3: Please show significant differences, similar to Fig. 1.

We have added the significant differences in Figure 3. It should be noted that these results were obtained by *in cellulo* assay (microspectrofluorimetry; Nat. Protoc. 2018; 10.1038/s41579-019-0294-2), which reflects the population heterogeneity where one point shows the response of one individual cell. Thus the different conditions were, for the most part, significantly very different.

13) Lines 214-216: By comparing NOR+ROS only to NOR and ROS, it seems hard to disentangle the direct effect ROS on bacterial killing and its indirect effect through competition. I think it would be useful to involve the NOR+CCCP result in the discussion, as this is supposed to indicate the maximal effect on killing that can be attained by competitive pump inhibition alone.

The result obtained with CCCP reflects the effect obtained in the absence of the pump energy driving force. We have used this condition as an accumulation control but the comparison of cell killings with CCCP seems delicate to us. Therefore, we have chosen to focus on the NOR+ROS comparison with NOR and ROS alone. We can see in figure 4 a killing level noticeably higher with NOR+ROS than the total of NOR killing + ROS killing.

REVIEWERS' COMMENTS:

Reviewer #3 (Remarks to the Author):

The authors have made substantial changes that improved the quality of the manuscript. They made several textual changes as well as inserted and deleted data, as requested by the reviewers. Hence, I believe my comments have been largely addressed, and I have no further comments.